# LIGHT SAMPLING FIELD AND BRDF REPRESENTATION FOR PHYSICALLY-BASED NEURAL RENDERING

**Jing Yang** *, **Hanyuan Xiao** *, **Wenbin Teng** , **Yunxuan Cai** , **Yajie Zhao**
Institute for Creative Technologies
University of Southern California
`{jyang,hxiao,wteng,ycai,zhao}@ict.usc.edu`

## ABSTRACT

Physically-based rendering (PBR) is key for immersive rendering effects used widely in the industry to showcase detailed realistic scenes from computer graphics assets. A well-known caveat is that producing the same is computationally heavy and relies on complex capture devices. Inspired by the success in quality and efficiency of recent volumetric neural rendering, we want to develop a physically-based neural shader to eliminate device dependency and significantly boost performance. However, no existing lighting and material models in the current neural rendering approaches can accurately represent the comprehensive lighting models and BRDFs properties required by the PBR process. Thus, this paper proposes a novel lighting representation that models direct and indirect light locally through a light sampling strategy in a learned light sampling field. We also propose BRDF models to separately represent surface/subsurface scattering details to enable complex objects such as translucent material (i.e., skin, jade). We then implement our proposed representations with an end-to-end physically-based neural face skin shader, which takes a standard face asset (i.e., geometry, albedo map, and normal map) and an HDRI for illumination as inputs and generates a photo-realistic rendering as output. Extensive experiments showcase the quality and efficiency of our PBR face skin shader, indicating the effectiveness of our proposed lighting and material representations.

## 1 INTRODUCTION

Physically-based rendering (PBR) provides a shading and rendering method to accurately represent how light interacts with objects in virtual 3D scenes. Whether working with a real-time rendering system in computer graphics or film production, employing a PBR process will facilitate the creation of images that look like they exist in the real world for a more immersive experience. Industrial PBR pipelines take the guesswork out of authoring surface attributes like transparency since their methodology and algorithms are based on physically accurate formulae and resemble real-world materials. This process relies on onerous artist tuning and high computational power in a long production cycle. In recent years, academia has shown incredible success using differentiable neural rendering in extensive tasks such as view synthesis (Mildenhall et al., 2020), inverse rendering (Zhang et al., 2021a), and geometry inference (Liu et al., 2019). Driven by the efficiency of neural rendering, a natural next step would be to marry neural rendering and PBR pipelines. However, none of the existing neural rendering representations supports the accuracy, expressiveness, and quality mandated by the industrial PBR process.

A PBR workflow models both specular reflections, which refers to light reflected off the surface, and diffusion or subsurface scattering, which describes the effects of light absorbed or scattered internally. Pioneering works of differentiable neural shaders such as Softras (Liu et al., 2019) adopted the Lambertian model as BRDF representation, which only models the diffusion effects and results in low-quality rendering. NeRF (Mildenhall et al., 2020) proposed a novel radiance field representation for realistic view-synthesis under an emit-absorb lighting transport assumption without explicitly modeling BRDFs or lighting, and hence is limited to a fixed static scene with no scope for relighting. In follow-up work, NeRV (Srinivasan et al., 2020) took one more step by explicitly modeling directional light, albedo, and visibility maps to make the fixed scene relightable. The indirect illumination was achieved by ray tracing under the assumption of one bounce of incoming light.

---

*Equal contributions. We would like to thank Marcel Ramos and Chinmay Chinara at Vision and Graphics Lab (VGL), for their valuable help on data preparation and paper writing.

However, this lighting model is computationally very heavy for real-world environment illumination when more than one incoming directional lights exist. To address this problem, NeRD (Boss et al., 2020) and PhySG (Zhang et al., 2021a) employ a low-cost global environment illumination modeling method using spherical gaussian (SG) to extract parameters from HDRIs. Neural-PIL (Boss et al., 2021) further proposed a pre-trained light encoding network for a more detailed global illumination representation. However, it is still a global illumination representation assuming the same value for the entire scene, which is not true in the real world, where illumination is subjected to shadows and indirect illumination bouncing off of objects in different locations in the scene. Thus it's still an approximation but not an accurate representation of the environmental illumination. Regarding material (BRDF) modeling, all the current works adopt the basic rendering parameters (such as albedo, roughness, and metalness) defined in the rendering software when preparing the synthetic training data. However, they will fail in modeling intricate real-world objects such as participating media (e.g., smoke, fog) and translucent material (organics, skins, jade), where high scattering and subsurface scattering cannot be ignored. Such objects require more effort and hence attract more interest in research in their traditional PBR process.

In this work, we aim to design accurate, efficient lighting/ illumination and BRDF representations to enable the neural PBR process, which will support high-quality and photo-realistic rendering in a fast and lightweight manner. To achieve this goal, we propose a novel lighting representation - a Light Sampling Field to model both the direct and indirect illumination from HDRI environment maps. Our Light Sampling Field representation faithfully captures the direct illumination (incoming from light sources) and indirect illumination (summary of all indirect incoming lighting from surroundings) given an arbitrary sampling location in a continuous field. Accordingly, we propose BRDF representations in the format of surface specular, surface diffuse, and subsurface scattering for modeling real-world object material. This paper mainly evaluates the proposed representations with a novel volumetric neural physically-based shader for human facial skin. We trained with an extensive high-quality database, including real captured ground truth images as well as synthetic images for illumination augmentation. We also introduce a novel way of integrating surface normals into volumetric rendering for higher fidelity. Coupled with proposed lighting and BRDFs models, our light transport module delivers pore-level realism in both on- and underneath-surface appearance unprecedentedly. Experiments show that our Light Sampling Field is robust enough to learn illumination by local geometry. Such an effect usually can only be modeled by ray tracing. Therefore, our method compromises neither efficiency nor quality with the Light Sampling Field when compared to ray tracing.

The main contributions of this paper are as follows: 1) A novel volumetric lighting representation that accurately encodes the direct and indirect illumination positionally and dynamically given an environment map. Our local representation enables efficient modeling of complicated shading effects such as inter-reflectance in neural rendering for the first time as far as we are aware. 2) A BRDF measurement representation that supports the PBR process by modeling specular, diffuse, and subsurface scattering separately. 3) A novel and lightweight neural PBR face shader that takes facial skin assets and environment maps (HDRIs) as input and efficiently renders photo-realistic, high-fidelity, and accurate images comparable to industrial traditional PBR pipelines such as Maya. Our face shader is trained with an image database consisting of extensive identities and illuminations. Once trained, our models will extract lighting models and BRDFs from input assets, which works well for novel subjects/ illumination maps. Experiments show that our PBR face shader significantly outperforms the state-of-the-art neural face rendering approaches with regard to quality and accuracy, which indicates the effectiveness of the proposed lighting and material representations.

## 2 RELATED WORK

**Volumetric Neural Rendering** Volumetric rendering models the light interactions with volume densities of absorbing, glowing, reflecting, and scattering materials (Max, 1995). A neural volumetric shader trains a model from a set of images and queries rendered novel images. The recent state-of-the-art was summarized in a survey (Tewari et al., 2020). In addition, Zhang et al. (2019) introduced the radiance field as a differentiable theory of radiative transfer. Neural Radiance Field (NeRF) (Mildenhall et al., 2020) further described scenes as differentiable neural representation and the following raycasting integrated color in terms of the transmittance factor, volume density, and the voxel diffuse color. Extensions to NeRF were developed for better image encoding (Yu et al., 2020), ray marching (Bi et al., 2020a), network efficiency (Lombardi et al., 2021; Yariv et al., 2020), realistic shading (Suhail et al., 2022) and volumetric radiative decomposition (Bi et al., 2020b; Rebain

et al., 2020; Zhang et al., 2021c; Verbin et al., 2021). In particular, NeRV (Srinivasan et al., 2020), NeRD (Boss et al., 2020; 2021) decomposed the reconstructed volume into geometry, SVBRDF, and illumination given a set of images even under varying lighting conditions. RNR (Chen et al., 2020b) assumed the environmental illumination as distant light and is able to decompose the scene into an albedo map with a 10-order spherical harmonics (SH) of incoming directions.

**Portrait and Face Relighting** Early single-image relighting techniques utilize CNN-based image translation (Nalbach et al., 2017; Thies et al., 2019). Due to the lack of 3D models, image translation approaches cannot recover surface materials or represent realistic high-fidelity details, thus neural volumetric relighting approaches are widely adopted recently. Ma et al. (2021) proposed a lightweight representation to decode only the visible pixels during rendering. Bi et al. (2021) The face relighting utilized strong priors. Zhou et al. (2019) fitted a 3D face model to the input image and obtained refined normal to help achieve relighting. Chen et al. (2020a) relit the image by using spherical harmonics lighting on a predicted 3D face. Hou et al. (2022) introduced a shadow mask estimation module to achieve novel face relighting with geometrically consistent shadows. With high-quality volumetric capture in lightstage (Debevec et al., 2000) to obtain training data, this trend achieved the following: regression of a one-light-at-a-time (OLAT) image for relighting (Meka et al., 2019), encoding the feature tensors for a Phong shading into UV space and relighting using an HDRI map (Meka et al., 2020), a neural renderer that can predict the non-diffuse residuals (Zhang et al., 2021b). Bi et al. (2021) proposed neural networks to learn relighting implicitly but lacked modeling both surface and subsurface reflectance properties following physical light transport. Similar to our approach most, Sun et al. (2021) inferred both light transport and density, and enabled relighting and view synthesis from a sparse set of input images.

## 3 METHODS

### 3.1 PRELIMINARIES

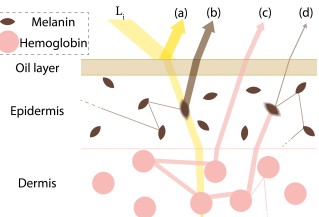

Figure 1: **Skin Scattering Model.** The incident light $L_i$ scattered from the skin consists of an oil layer specular reflection (a) or penetrates one or more scattering layers (b, c, d) at some point underneath the oil layer. The difference in melanin distribution or color results in the diversity of skin appearance. In humans, the distribution and color of melanin is the primary determinant of skin appearance. Darker and denser melanin in the epidermis leads to darker skin color and vice versa.

We use the rendering equation Kajiya (1986) to estimate the radiance $L$ at a 3D point $\mathbf{x}$ with the outgoing direction $\omega_o$: $L(\mathbf{x}, \boldsymbol{\omega}_o) = \int_{\boldsymbol{\omega}_i \in \Omega^+} f(\mathbf{x}, \boldsymbol{\omega}_o, \boldsymbol{\omega}_i) L_i(\mathbf{x}, \boldsymbol{\omega}_i)(\boldsymbol{\omega}_i \cdot \boldsymbol{n}_x) d\boldsymbol{\omega}_i$, where $f(\mathbf{x}, \boldsymbol{\omega}_o, \boldsymbol{\omega}_i)$ is the BRDF representation and $L_i(\mathbf{x}, \boldsymbol{\omega_i})$ measures the radiance of incident light with direction $\boldsymbol{\omega_i}$.

Generally, the incident light could be categorized as direct vs. indirect light. Fig. 1 illustrates an example of direct light (path (a), (b)) and indirect light (path (c)) in human face skin, where subsurface scattering happens in the deeper dermis layer, causing the nearby area to receive indirect light. Therefore, the rendering formulation can be split into separate components with direct and indirect lighting:

$$L(\mathbf{x}, \boldsymbol{\omega}_o) = \int_{\boldsymbol{\omega}_i \in \Omega^+} f_s(\mathbf{x}, \boldsymbol{\omega}_o, \boldsymbol{\omega}_i) L_i^d(\mathbf{x}, \boldsymbol{\omega}_i)(\boldsymbol{\omega}_i \cdot \boldsymbol{n}_x) d\boldsymbol{\omega}_i + \int_{\boldsymbol{\omega}_i \in \Omega} f_{ss}(\mathbf{x}, \boldsymbol{\omega}_o, \boldsymbol{\omega}_i) L_i^{id}(\mathbf{x}, \boldsymbol{\omega}_i)(\boldsymbol{\omega}_i \cdot \boldsymbol{n}_x) d\boldsymbol{\omega}_i \quad (1)$$

where $f_s$, $f_{ss}$ represent the BRDF evaluation of surface and subsurface, respectively. Following the render equation, we design a lightweight physically-based rendering method by learning different BRDF representations and modeling the direct and indirect lights, as we will introduce in the next few sections.

### 3.2 ILLUMINATION AND BIDIRECTIONAL REFLECTANCE DISTRIBUTION LEARNING

We propose a lighting model and material model to construct the Light Sampling Field and estimate the BRDF representations, as will be introduced in the next several sections.

#### 3.2.1 LIGHTING MODEL

Considering the interaction between light and different skin layers, we propose a novel light modeling method using HDRIs that decomposes lighting into direct illumination and indirect illumination.

For direct illumination, We use light importance sampling to simulate external light sources, such as light bulbs. And we implement ray tracing for the specular reflectance effect. For indirect illumination, we introduce a Light Sampling Field that models location-aware illumination using SH. This learned local light-probe models subsurface scattering and inter-reflectance effects.

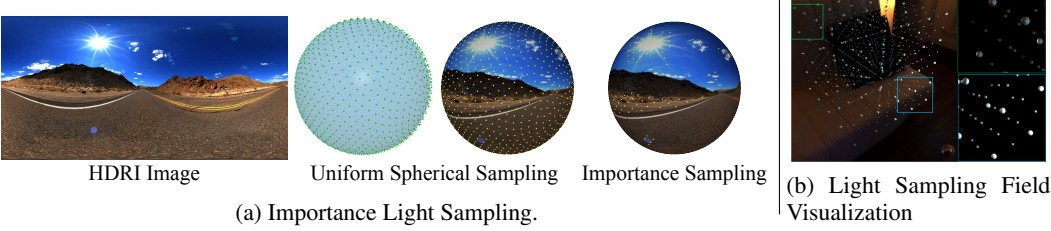

(a) Importance Light Sampling.

(b) Light Sampling Field Visualization

Figure 2: **Lighting Model.** (a) We use importance sampling to sample heavily on pixels with high intensity. (b) We use various densities of sampling to bring attention to locations with data as in the Light Sampling Field.

**Importance Light Sampling for Direct Illumination.** Direct radiance comes from the HDRI map directly. In order to compute the contribution of pixels in HDRI to different points in our radiance field, we use a SkyDome to represent direct illumination by projecting the HDRI environment map onto a sphere. Each pixel on the sphere is regarded as a distant directional light source. Hence, direct lighting is identical at all locations in the radiance field. Such representation preserves specular reflection usually achieved by ray tracing methods. We take two steps to construct the representation. Firstly, we uniformly sample a point grid of size $N = 800$ on the sphere, where each point is a light source candidate. Secondly, we apply our importance light sampling method to filter valid candidates by two thresholds: 1) an intensity threshold that clips the intensity of outliers with extreme values; 2) an importance threshold that filters out outliers in textureless regions. Fig. 2a illustrates this process.

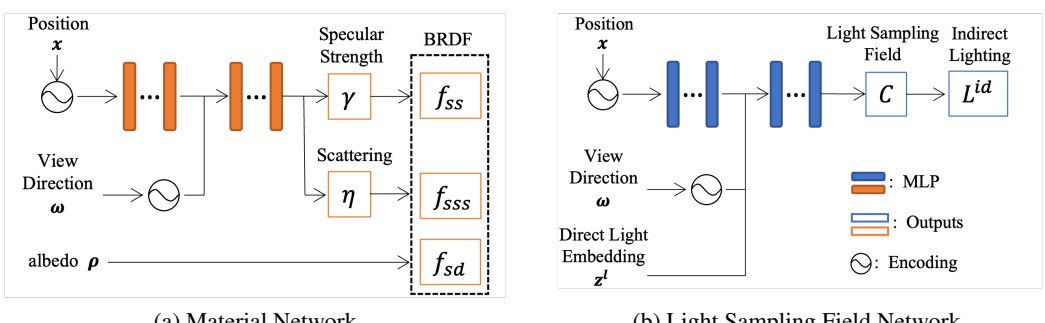

(a) Material Network

(b) Light Sampling Field Network

Figure 3: **Material and Light Sampling Field Network.** (a) Our material network takes in 3D position and view direction as inputs and predicts the specular strength and skin scattering parameters of our BRDF model. (b) The Light Sampling Field network applies similar differentiable network but with the light sampling created by our importance light sampling technique as another input to predict SH coefficients.

**Light Sampling Field for Indirect Illumination.** Indirect illumination models the incoming lights reflected or emitted from surrounding objects, which is achieved by ray tracing with the assumption of limited bounce times in the traditional PBR pipeline. Inspired by the volumetric lightmaps used in Unreal Engine (Karis & Games, 2013), which stores precomputed lighting in sampled points and use them for interpolation at runtime for modeling indirect lighting of dynamic and moving objects, We adopt a continuous Light Sampling Field for accurately modeling the illumination variation in different scene positions. We use Spherical Harmonics (SH) to model the total incoming lights at each sampled location separately. We compute the local SH by multiplying fixed Laplace's SH basis with predicted SH parameters. Specifically, we use SH of degree $l = 1$ for each color channel (RGB). Therefore, we acquire 3 (color channels) $\times$ 4 (basis) = 12-D vector as local SH representation. We downsample the HDRI map to $100 \times 150$ resolution and project it to a sphere. Each pixel on the map is considered an input lighting source. We use the direction and color of each pixel as the lighting embedding to feed into a light field sampling network for inference of

coefficients of local SH. We visualize our Light Sampling Field with selected discrete sample points in Fig. 2b.

**Learning Light Sampling Field.** We design a network (Fig. 3b) to predict the spherical Harmonics coefficient $C_k^m$ of a continuous light field. The inputs of this network are the lighting embedding $z^l$, the positional encoding of 3D location $\mathbf{x}$, and view direction $\boldsymbol{\omega}$. Conditioned on the lighting representations, the network succeeds in predicting the accurate and location-aware lighting. Fig. 6b evaluates our lighting model.

### 3.2.2 MATERIAL MODEL

The choice of reflectance parameters usually relies on artists' tuning and therefore requires high computational power and results in a long production cycle, but sometimes not very ideal. To tackle the problem, we propose a lightweight material network (Fig. 3a) to estimate the BRDF parameters including specular strength $\gamma \in \mathbb{R}$ and skin scattering $\eta \in \mathbb{R}^3$ through learnable parameters. The parameters are crucial to represent the reflectance property of both surface and subsurface. Together with input albedo $\rho$, we can construct a comprehensive BRDF to model surface reflection and subsurface scattering. It consists of a surface specular component $f_{ss}$, a surface diffuse component $f_{sd}$, and a subsurface scattering component $f_{sss}$. Refer to Sec. 3.3 for a detailed light transport and Fig.5 for an evaluation of modeling subsurface scattering.

### 3.3 LIGHT TRANSPORT

Light transport defines a light path from the luminaire to the receiver. We introduce the light transport that connects our lighting model and material model in Fig. 4. We also detail the rendering equation in this section to match the light transport along the light path.

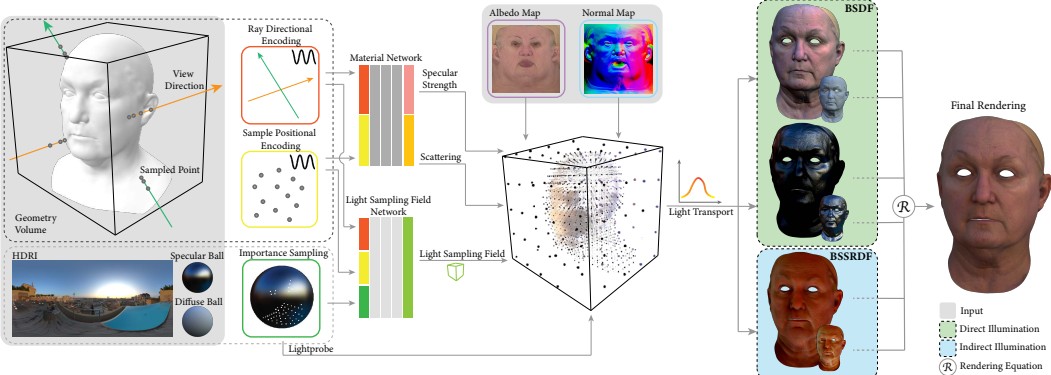

Figure 4: **Light Transport.** Starting from sampling lights from HDRI and 3D points close to geometry along the light path, we feed encoded locations, view direction to Material network and Light Sampling Field network (with extra direct illumination code). The networks output material and indirect illumination code. Our PBR equation generates radiance at each sample based on our material model and light transport in the volume and further composites all the radiance on the light path to the receiver.

**Volume Casting.** Our lighting model includes direct illumination and indirect illumination. Light transports from explicit lights and casts along the light path. We, therefore, define the light transport along the light path, $\boldsymbol{p}$, in the volume by the following:

$$L(\hat{\mathbf{x}}, \boldsymbol{\omega}_o) = \int_0^\infty \tau(t) \cdot \sigma(\mathbf{p}(t)) \cdot L(\mathbf{p}(t), \boldsymbol{\omega}_o) \, dt \tag{2}$$

where $L(\hat{\mathbf{x}}, \boldsymbol{\omega}_o)$ is the total radiance at $\hat{\mathbf{x}}$ along the light path direction $\boldsymbol{\omega}_o$. $\sigma(\mathbf{x})$ is the volume density, which is converted from the input geometry. $\tau(x) = \exp(-\int_0^x \sigma(\boldsymbol{p}(t))dt)$ is the visibility that indicates whether the location $x$ is visible on the light path, where $\boldsymbol{p}(t)$ represents the 3D location on the light path along $\boldsymbol{\omega}_o$ from $\hat{\mathbf{x}}$ and defined as $\boldsymbol{p}(t) = \hat{\mathbf{x}} + t\boldsymbol{\omega}_o$.

We use explicit geometry to construct a more reliable density field. More specifically, we locate the intersection coordinates, $\mathbf{x_0}$, between any arbitrary light path and the input geometry. For any point $\mathbf{x}$ along the light path $\boldsymbol{\omega}$, we define the density $\sigma$ as the following central Gaussian distribution

function: $\sigma(\mathbf{x}) = \alpha_\sigma \cdot \exp\left(-d_G(\mathbf{x})^2/(2\delta^2)\right)$, where $d_G(\mathbf{x})$ is the distance between $\mathbf{x}$ and the intersection $\mathbf{x_0}$, $\alpha_\sigma$ and $\delta$ are two scalars that determine the magnitude and standard deviation of the gaussian distribution.

**Material Scattering.** The light transport between the scene objects and light sources is characterized by the rendering equation. Eqn. 1 introduce the rendering covering direct illumination and indirect illumination. Also, classified by reflection location, the reflected radiance $L(\mathbf{x}, \boldsymbol{\omega}_o)$ has two components: surface reflectance and subsurface volume scattering. To have a comprehensive light transport representation, we further develop the equation with the dissection of light as well as the specialized BSSRDF components:

$$L(\mathbf{x}, \boldsymbol{\omega}_o) = \underbrace{\int_{\boldsymbol{\omega}_i \in \Omega_+} f_{ss}(\mathbf{x}, \boldsymbol{\omega}_o, \boldsymbol{\omega}_i) L_i^d(\mathbf{x}, \boldsymbol{\omega}_i)|\boldsymbol{\omega}_i \cdot \mathbf{n_x}|d\boldsymbol{\omega}_i}_{\text{surface specular reflectance with direct light}} + \underbrace{\int_{\boldsymbol{\omega}_i \in \Omega_+} f_{sd}(\mathbf{x}, \boldsymbol{\omega}_o, \boldsymbol{\omega}_i) L_i^d(\mathbf{x}, \boldsymbol{\omega}_i)|\boldsymbol{\omega}_i \cdot \mathbf{n_x}|d\boldsymbol{\omega}_i}_{\text{surface diffuse reflectance with direct light}}$$

$$+ \underbrace{\int_{\boldsymbol{\omega}_i \in \Omega} f_{sss}(\mathbf{x}, \boldsymbol{\omega}_o, \boldsymbol{\omega}_i) L_i^{id}(\mathbf{x}, \boldsymbol{\omega}_i)|\boldsymbol{\omega}_i \cdot \mathbf{n_x}|d\boldsymbol{\omega}_i}_{\text{subsurface scattering with indirect light}} \tag{3}$$

here $L_i^d(\mathbf{x}, \boldsymbol{\omega}_i)$ and $L_i^{id}(\mathbf{x}, \boldsymbol{\omega}_i)$ are the incoming direct and indirect radiance from direction $\boldsymbol{\omega}_i$ at point $\mathbf{x}$, respectively. $f_{ss}$, $f_{sd}$, and $f_{sss}$ are the different counterparts of light transport parameterized by material representations. We show the complete light transport in Eqn. 4 consists of our light and material representation.

$$L(\mathbf{x}, \boldsymbol{\omega}_o) = \underbrace{\gamma_{\mathbf{x}} \cdot \int_{\boldsymbol{\omega}_i \in \Omega_+} L_i^d(\mathbf{x}, \boldsymbol{\omega}_i)|\boldsymbol{\omega}_o \cdot R(\boldsymbol{\omega}_i, \mathbf{n_x})|^e d\boldsymbol{\omega}_i}_{\text{surface specular reflectance with direct light}} + \underbrace{\frac{\rho_{\mathbf{x}}^s}{\pi} \cdot \int_{\boldsymbol{\omega}_i \in \Omega_+} L_i^d(\mathbf{x}, \boldsymbol{\omega}_i)|\boldsymbol{\omega}_i \cdot \mathbf{n_x}|d\boldsymbol{\omega}_i}_{\text{surface diffuse reflectance with direct light}}$$

$$+ \underbrace{\frac{\rho_{\mathbf{x}}^{ss} + \eta_{\mathbf{x}}}{\pi} \cdot \int_{\boldsymbol{\omega}_i \in \Omega} L_i^{id}(\mathbf{x}, \boldsymbol{\omega}_i)|\boldsymbol{\omega}_i \cdot \mathbf{n_x}|d\boldsymbol{\omega}_i}_{\text{subsurface scattering with indirect light}} \tag{4}$$

where $\gamma_{\mathbf{x}}$ and $\eta_{\mathbf{x}}$ are the predicted specular strength and scattering from material network at $\mathbf{x}$. $R(\boldsymbol{\omega}_i, \mathbf{n_x})$ denotes the reflection direction of $\boldsymbol{\omega}_i$ at the surface with normal $\mathbf{n_x}$ and $e$ denotes the specular exponent. Also, $\rho_{\mathbf{x}}^s$ and $\rho_{\mathbf{x}}^{ss}$ are surface and subsurface albedo at $\mathbf{x}$ sampled from the input albedo map correspondingly with geometry.

## 4 IMPLEMENTATION DETAILS

In order to construct the density field $\sigma$, we set $\alpha_\sigma$ and $\delta$ to be 10 and 0.5, respectively. We further compare the rendering results of other values and visualize them in the Appendix. In the constructed radiance field, to sample rays, we draw 1024 random rays per batch. Along each ray, we sample 64 points for the shading model. The low-frequency location of 3D points and direction of rays are transformed to high-frequency input via positional encoding and directional encoding respectively (Mildenhall et al., 2020). The length of encoded position and view direction is 37 and 63 respectively in the material network and the Light Sampling Field network. Also, importance light sampling takes 800 light samples $z \in \mathbb{R}^3$ from the HDRI input for direct lighting. We further downsample the input HDRI and embedded all pixels as a light embedding $z^l \in \mathbb{R}^{6 \times 15000}$ to feed into the Light Sampling Field network. We use an 8-layer MLP with 256 neurons in each layer for both networks. For the material network, encoded sample locations are fed in the first layer of the MLP, while the encoded view direction is later fed in layer 4. The output of material MLP for each queried 3D point is specular strength $\gamma \in \mathbb{R}$, and scattering $\eta \in \mathbb{R}^3$. The Light Sampling Field network has a similar network structure but also has direct light embedding in layer 4 as input and outputs encoded spherical coefficients $C_k^m \in \mathbb{R}^{12}$ for indirect lighting. During light transport, we obtain a weighted value for each ray based on $\tau$ distribution among the sampled points along the ray. After introducing values from pre-processed albedo and normal maps, the value of each component on rays is gathered and visualized as an image with pixels representing their intensities following our rendering Eqn. 4. Finally, the rendered RGB values are constrained with ground truth by an MSE loss. In our application, MLP modules can converge in $50,000$ iterations (2.6 hour) on a single Tesla V100, with decent results on the same level of detail as reference images.

# 5 EXPERIMENTS AND EVALUATION

## 5.1 TRAINING DATASET

Our training dataset is composed of a synthetic image dataset and a Lightstage-scanned image dataset. In *synthetic dataset*, we used a professionally-tuned Maya face shader to render 40-view colored images under all combinations between 21 face assets and 101 HDRI+86 OLAT illumination. *Lightstage-scan dataset* consists of 16-view captured colored images of 48 subjects in 27 expressions under white illumination. We carefully selected subjects in both dataset preparation to cover diverse ages, skin colors, and gender. Further details can be found in Appendix. A.

## 5.2 EVALUATION AND ANALYSIS

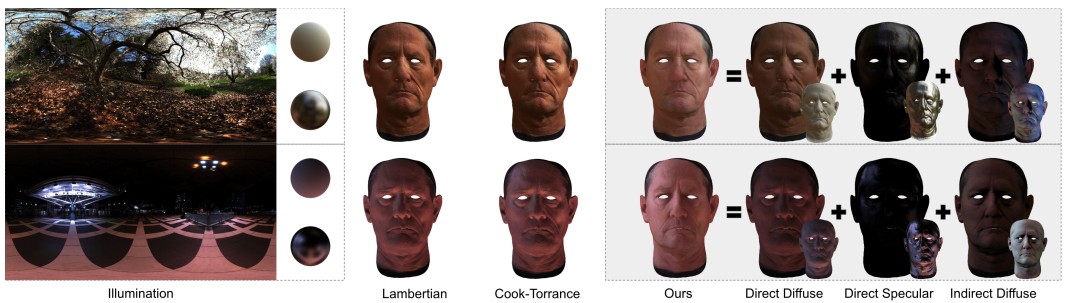

Figure 5: **Material Modeling.** Under HDRI illumination in column (a), we show the appearance of (b) Lambertian BRDF, (c) Cook-Torrance BRDF, and (d) ours (including direct diffuse, direct specular, and indirect diffuse). We adjust the intensity of each component for visualization purposes.

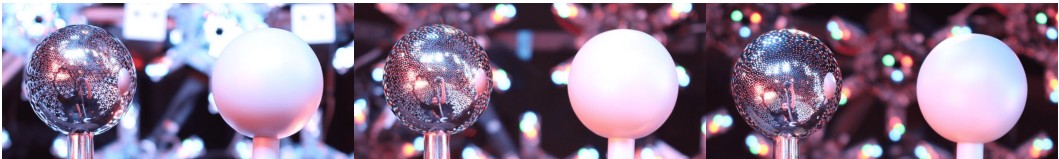

(a) Appearance at three different locations under the same illumination

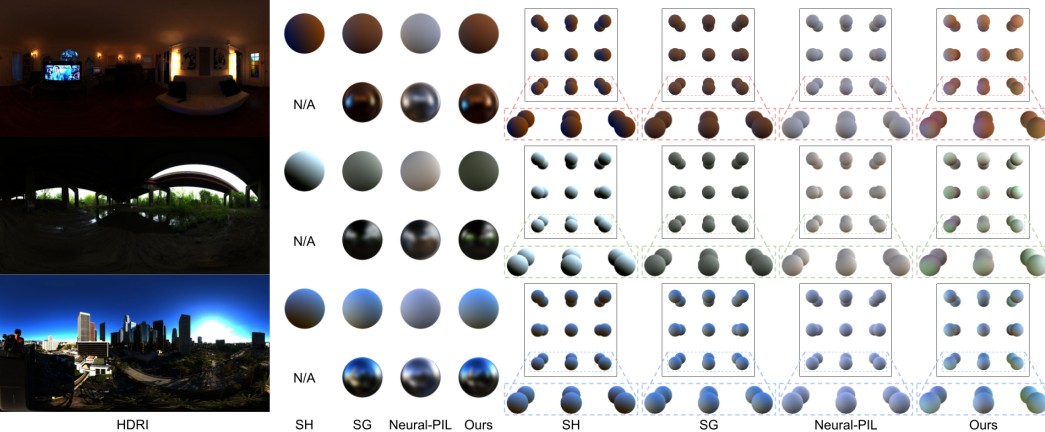

(b) Lighting Model Evaluation.

Figure 6: Light Evaluation. (a) Given HDRI illumination, we capture the appearance of a mirror ball and a grey ball in the Light Stage at different locations. (b) We rendered a uniform white ball and a grid of such balls with different lighting models: SH (degree $l = 1$), SG, Neural-PIL, and ours. We separated diffuse lighting and specular lighting by turning off the corresponding component in the rendering equation.

**Material Model Evaluation.** We conduct a comparison experiment with only surface scattering (BRDF) in Fig. 5, which presents two BRDF materials (middle two columns) and our proposed material with layer-by-layer decomposition (right four columns). From the comparison, Cook-Torrance BRDF has a more specular effect than Lambertian yet saves rubber-alike appearance from the uncanny valley. Apart from surface scattering, our method also predicts subsurface scattering to achieve a vivid look around the nose tip and ears by depicting red blood cells and vessel color.

**Lighting Model Evaluation.** We show our captured real data under a real fixed HDRI illumination in Fig. 6a. Fig. 6b illustrates uniform white ball illumination within a scene rendered by SH lighting (degree $l = 1$), Spherical Gaussian (SG) Neural-PIL(Boss et al., 2021) and our method, respectively. Compared with other models, our proposed method delivers the highest fidelity of illumination with the widest spectrum of light as well as a lighting field sensitive to lighting distribution. We further provide an extensive ablation study to validate our light sampling for modeling direct illumination in Fig. 15.

**Evaluation of Light and Material Modeling in indirect illumination.** We evaluate our light and material components in Fig. 7. In (c), we use pre-calculated SH of degree $l = 1$ to model the global illumination and albedo as a diffuse scattering to render the face, resulting in a face image with strong shadows and an unnatural appearance. In (b), we introduce learned subsurface scattering in material but still use the same pre-calculated SH of degree $l = 1$ as global illumination, which results in color shift and artifacts. In (c), we further introduce local SH and infer a light sampling field to replace the pre-calculated SH. Together with the full spectrum of material layers, we achieve realistic rendering effects. In particular, we demonstrate interreflection effects in the zoom-in box. The shadow is softened by modeling the scattering and positional illumination.

Figure 7: **Evaluation of light and material modeling.** (a) Rendering results using proposed materials and location-aware local SH as lighting model. (b) Rendering results using proposed materials with global SH. (c) Rendering results with global SH and without using the subsurface scattering layer in the material. We also demonstrate the inter-reflection effects in the zoom-in boxes. Compare to (c) and (b), (a) achieves soft shadows.

**Inverse Rendering.** Our method can also achieve high-fidelity inverse rendering with multi-view images (under various illumination) instead of geometry and texture maps as input. To make this possible, we additionally implemented MLP to predict a density field. We present our results under novel illumination in Fig. 8.

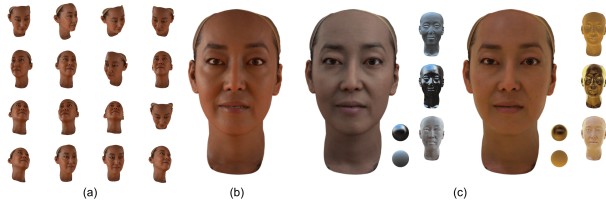

Figure 8: **Inverse Rendering.** (a) An example input of a set of multi-view images. (b) Reference view under the input illumination. (c) Inverse rendering results in two novel environment illuminations.

## 5.3 QUALITATIVE RESULTS

**Rendering on General Object Assets.** We picked orange and meat in addition to face subjects to show that our method is generalizable on diverse objects with multi-layered structures in Fig. 9. Through testing on different organic materials, we show consistent sharpness and realistic appearance, especially in accomplishing specular reflection on orange and accurate soft shadow on meat.

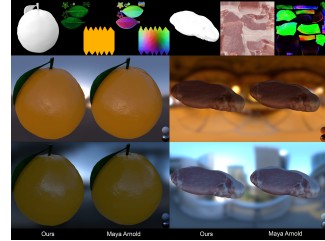

Figure 9: **General objects.**

**Maya Comparison.** We compare our renderings with Maya, an industry-level rendering engine, under HDRI or OLAT (one-light-at-a-time) illumination and present zoom-in pore-level details in Fig. 10. The zoom-in inspections show comparable or even better

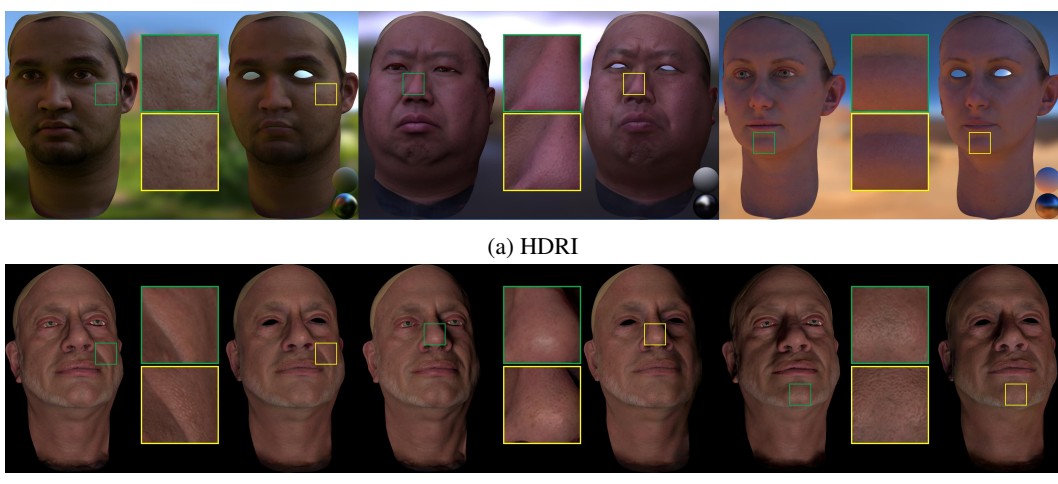

(a) HDRI

(b) OLAT

Figure 10: Qualitative comparison between ours and Maya renderings with zoom-in under (a) HDRI and (b) OLAT. The left images and green crops in each pair are Maya renderings. The right images and yellow crops in each pair are ours. Our result achieves a realistic appearance of skin texture in addition to accurate diffuse and ambient illumination effects.

rendering results in their sharpness or illumination. Skin wrinkles and forehead specularity are particularly rich and sharp from the proposed method. At testing time, with the same rendering assets and queries ($2500 \times 2500$ in resolution), our method requires the training images with only $800 \times 800$ in resolution. Under OLAT settings, our method casts hard shadows and soft shadows as accurately as Maya. More comparisons and qualitative results can be found in Fig. 17 and Fig. 18.

**Qualitative Comparison.** In Fig. 16, we compare the rendering results of FRF (Tewari et al., 2021), NeLF (Sun et al., 2021), SIPR (Sun et al., 2019), Neural-PIL (Boss et al., 2021), and our methods under HDRI and OLAT illuminations. We present more testing performance of our trained-once models on other novel subjects in other datasets in Fig.19.

### 5.4 QUANTITATIVE RESULTS

We evaluated PSNR, SSIM, and LPIPS in quantitative measurements (Table 1) with Maya rendering as a benchmark. Specifically, LPIPS evaluates the perception on the VGG-19 network. Ours outperforms other baseline methods in all three metrics. Moreover, We do not provide the SSIM score of NeLF due to the slight view difference.

| Baseline | FRF | SIPR | NeLF | Neural-PIL | Ours |
|---|---|---|---|---|---|
| PSNR ↑ | 29.83 | 33.43 | 33.80 | 33.22 | **36.69** |
| SSIM ↑ | 0.4489 | 0.8507 | N/A | 0.9153 | **0.9250** |
| LPIPS (VGG) ↓ | 0.6017 | 0.1265 | 0.7538 | 0.1082 | **0.0664** |

Table 1: **Facial rendering metrics at** $800 \times 800$ **resolution.**

### 6 CONCLUSION

We demonstrate that the prior neural rendering representation for physically-based rendering fails to accurately model environment lighting or capture subsurface details. In this work, we propose a differentiable light sampling field network that models dynamic illumination and indirect lighting in a lightweight manner. In addition, we propose a flexible material network that models subsurface scattering for complicated materials such as the human face. Experiments on both synthetic and real-world datasets demonstrate that our light sampling field and material network collectively improve the rendering quality under complicated illumination compared with prior works. In the future, we will focus on modeling more complicated materials such as translucent materials and participated media. We will also collect datasets based on general objects and apply them for extensive tasks such as inverse rendering.

## 7 ACKNOWLEDGMENT

This research is sponsored by the U.S. Army Research Laboratory (ARL) under contract number W911NF-14-D-0005. Army Research Office also sponsored this research under Cooperative Agreement Number W911NF-20-2-0053. We would also would like to acknowledge Sony Corporation of America R&D Center, US Lab for their support. Statements and opinions expressed, and content included, do not necessarily reflect the position or the policy of the Government, and no official endorsement should be inferred. Further, the views and conclusions contained in this document are those of the authors and should not be interpreted as representing the official policies, either expressed or implied, of the Army Research Office or the U.S. Government. The U.S. Government is authorized to reproduce and distribute reprints for Government purposes notwithstanding any copyright notation herein.

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

# A  DATASETS

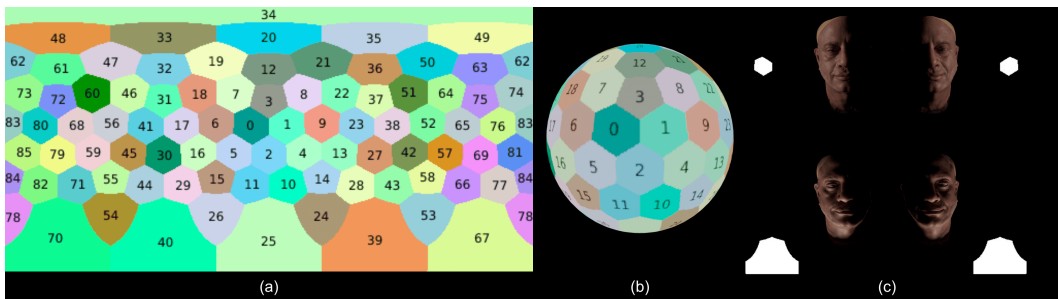

Figure 11: **OLAT Mapping.** (a) Area labels for OLAT mapping; (b) Spherical mapping; (c) OLAT mapping and renderings.

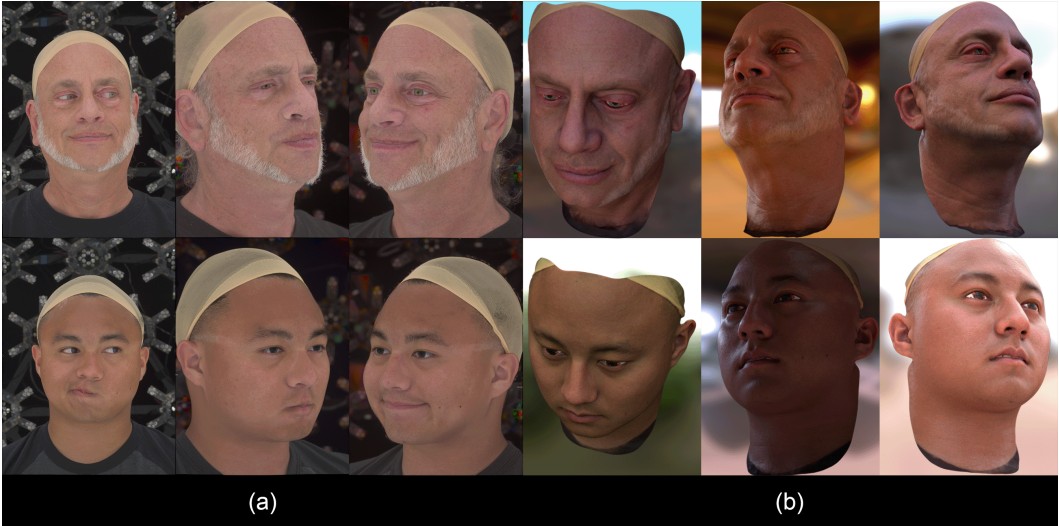

Figure 12: **Training Dataset.** (a) Light Stage-scanned real image dataset, (b) Maya-rendered synthetic image dataset

## A.1  LIGHT STAGE

**Face Capture.**  We use the Light Stage (Debevec, 2012) to capture data for training purposes. The Light Stage features controllable lights and cameras, allowing us to capture multiview images using polarized spherical gradient illumination (Ma et al., 2007; Ghosh et al., 2011). By decomposing the specular and diffuse surface reflections from the captured images, we can generate image space diffuse albedo and high-frequency normals. We further fit the reconstructed 3D face to our template mesh for consistent UV space textures.

**OLAT Mapping.**  To simulate OLAT illumination, we approximate a directional light source by using an area light source positioned on a sphere. We then convert this area light into a High Dynamic Range Image (HDRI) map using equirectangular mapping. We illustrate the segmentations and corresponding OLAT mapping in Fig. 11.

## A.2  TRAINING DATASET

Our training dataset consists of 1) a Light Stage-scanned multi-view colored image dataset under white illumination shown in Fig. 12 (a), 2) a synthetic multi-view colored image dataset rendered via a professionally-tuned Maya face shader shown in Fig. 12 (b). In the following sections, we introduce data composition and settings for synthetic data rendering and lightstage scanning.

**Synthetic Image Dataset**    Our input rendering assets to Maya renderer are 101 HDRI environment maps, 86 OLAT environment maps, and 21 face assets: the HDRI data covers various illuminations from outdoor open areas to small indoor areas; OLAT environment map cover 86 different regions in SkyDome as directional light; 21 subjects cover a variety of skin color, age, and gender. Each face asset consists of a coarse mesh, an albedo map, and a high-frequency normal map. We rendered 40 fixed-view RGBA-space images under all combinations of illumination and face assets. In total, we acquired $37,160$ images in $800 \times 800$-pixel resolution for our synthetic image dataset.

**Lightstage-scanned Real Image Dataset**    Synthetic data is insufficient to train a model with output rendering close to the real world. We utilized lightstage to additionally capture multi-view images for 48 subjects under uniform white illumination. Besides the diverse skin colors, gender, and age of subjects, the lightstage-scanned dataset consists of 27 expressions of each subject. We set up 16 fixed cameras covering different viewpoints of the frontal view. In total, we acquired $20,736$ images in $3008 \times 4096$-pixel resolution for our real image dataset.

## B    MORE RESULTS AND ANALYSIS

**Density Field Construction.**    We show how $\alpha_\sigma$ and $\delta$ affect the constructed density field in Fig. 13. Larger $\delta$ results in a more evenly distributed density at samples along the light path. The constructed density field hence presents a coarse boundary around the input geometry. Therefore, the rendered results tend to have a blurry appearance. However, smaller $\delta$ approaches Dirac delta distribution, where only samples close to the intersections have valid density values. The rendered results thus have black stripes. Larger $\alpha_\sigma$ alleviates the false density construction and results in accurate density values around the input geometry.

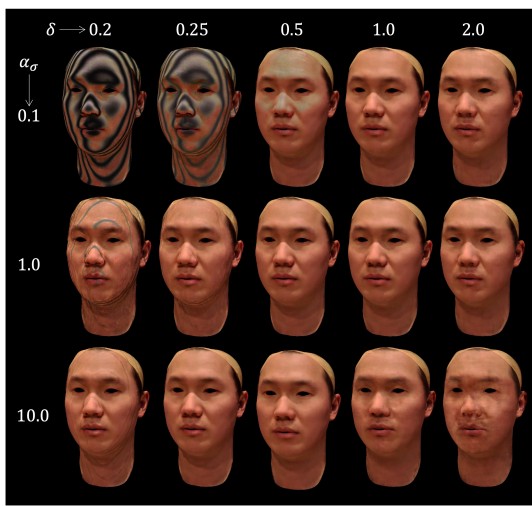

Figure 13: **Desnity Field Construction.**    We present a comparison of hyperparameters ($\alpha_\sigma$ and $\delta$) in density field construction.

**High-frequency Surface Normal.**    To investigate the effectiveness of high-frequency surface normal as input, we compare results by replacing high-frequency input normal maps with a low-frequency one extracted from coarse geometry (Fig. 14). Given other input assets unchanged, rendered results with low-frequency normal maps show the incapability of extracting high-frequency information from input images. Therefore, high-frequency normal maps are crucial in achieving pore-level sharpness even though training data contains high-frequency details on images.

**Ablation Study on Light Sampling**    We validate the Importance Light Sampling for simulating various external light sources with an extensive ablation study in Fig. 15. Under the same illumination conditions, the Importance Light Sampling can generate soft and appropriate diffuse reflection while preserving the accurate lighting distribution from the input HDRI. It ensures that the rendered images maintain the high degree of realism and fidelity to the original lighting conditions. In contrast, Uniform Spherical Sampling, while capable of representing the lighting environment with the same number of sampled lights, tends to produce hard shadows and may result in less detailed and overexposed images.

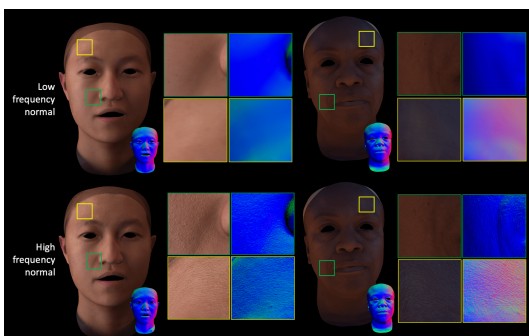

Figure 14: High-frequency Surface Normal

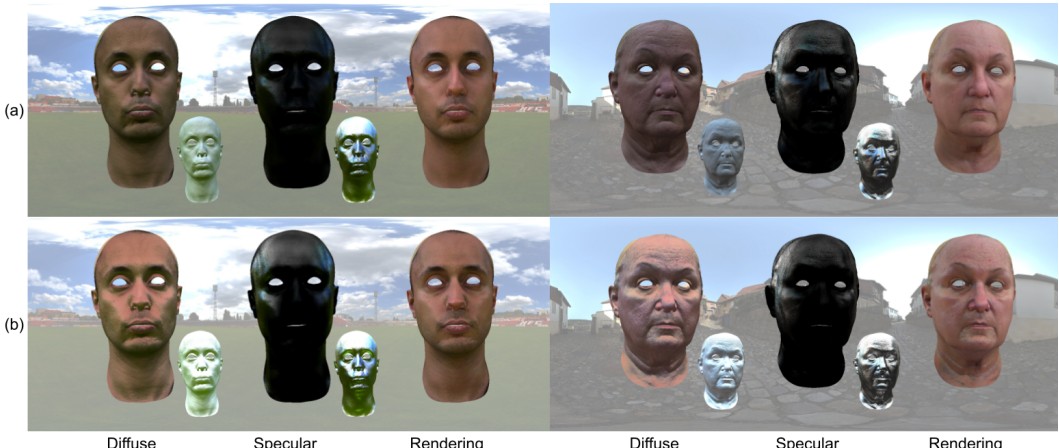

Figure 15: **Ablation study on the impact of different light sampling techniques.** We showcase the layer-by-layer decomposition of the direct lighting model using (a) Importance Light Sampling, and (b) Uniform Spherical Sampling. We adjust the intensity of each component equally for visualization purposes.

| Methods | Input | | Output | | Novel Subjects | Novel Viewpoint |
|---------|-------|-------|--------|------|----------------|-----------------|
| | Views | Known Lighting | Directional Light | HDRI | | |
| SIPR | Single-view | SH | | | ✓ | |
| FRF | Single-view | OLAT | ✓ | ✓ | ✓ | |
| Neural-PIL | Single-view | OLAT | ✓ | ✓ | ✓ | ✓ |
| NeLF | Multi-view | | | ✓ | | ✓ |
| Ours | Multi-view* | HDRI | ✓ | ✓ | ✓ | ✓ |

Table 2: **Input, output, and functionality of our baselines.** Multi-view* indicates that we first convert geometry and textures to multiview representation utilizing the ray-mesh intersection. Our method is capable of rendering novel subjects from novel viewpoints using either directional light or HDRI as queried illumination, without compromising on quality.

**Qualitative Comparison.** In Fig. 16, we compare the rendering results of FRF, NeLF, SIPR, Neural-PIL, and our methods under HDRI and OLAT (one-light-at-a-time) illuminations, as well as using multi-view images as input instead of geometry. To be more concrete, we identify different methods' input, output, and functionality in Table 2. Our method stands out for its ability to produce clear hard shadows resulting from full occlusion by face geometry, and soft shadows caused by indirect illumination. In contrast, Neural-PIL and NeLF do not model directional light and were not trained on OLAT data, so we compare them only under HDRI illumination. SIPR is an image-based relighting method that models the scene in 2D, and cannot be queried from novel viewpoints. FRF, Neural-PIL, and NeLF, on the other hand, models in 3D. Neural-PIL inherits a per-scene-per-train manner as NeRF and, thus is not generalizable on different subjects. In addition, we provide rendering results from Maya, a top-notch industrial renderer, as a reference for comparison.

**More Qualitative Results.** Fig. 18 presents a comprehensive collection of qualitative results. Each row showcases the rendering inputs, including the geometry, albedo map, and normal map, followed by six rendered images. The first three show different facial expressions under all-white illumination, while the last three display neutral expressions under different lighting conditions. By utilizing our photo-realistic neural renderer, we are able to render images at any resolution without compromising quality.

**Rendering Speed.** In addition to the rendering results, we also compare the rendering speed in Table 3. With the same specification of output and environment, our method is able to achieve up to 47-49 times faster with engineering acceleration (e.g. multi-thread processing)

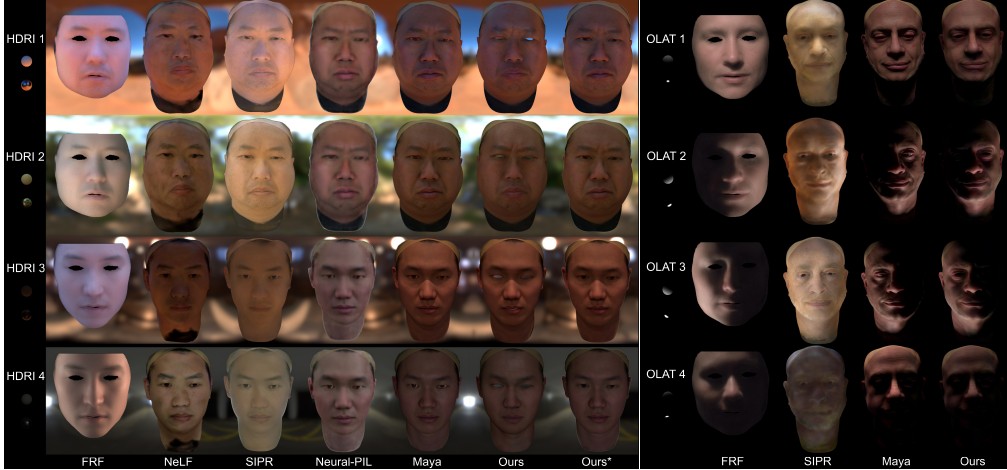

    (a) HDRI illumination                      (b) OLAT illumination

Figure 16: **Qualitative comparison with baseline methods.** (a) Under HDRI illumination, our rendering results achieve the highest sharpness and less plastic/rubber-like appearance, where *Ours* takes geometry and textures as input and *Ours\** takes multiview inputs; (b) OLAT illumination simulates strong directional lighting. While our robust direct light modeling enables hard shadow casting, our indirect light modeling realistically softens shadow at the boundary.

| Resolution | $800 \times 800$ | $1440 \times 1440$ | $2880 \times 2880$ | $5760 \times 5760$ |
|---|---|---|---|---|
| Maya | 130s | 370s | 1857s | 7848s |
| Ours | **3s** | **8s** | **31s** | **158s** |

Table 3: Facial rendering speed benchmark on different resolution in seconds

**Testing on Other Datasets.** To demonstrate that high-quality training data does not determine the robustness of the method, we evaluated our trained model over other available resources in Fig. 19. We converted three datasets to match our input as follows,

- *Triplegangers.* We used FaceX 3DMM to fit and align geometries; albedo map was directly transferred from source; normal map was inferred using (Wang et al., 2018).
- *3D Scan Store.* We used 3DMM to fit and align geometries; albedo and normal maps were provided and not further processed.
- *FaceScape.* Geometries and albedo maps were provided and not further processed. The normal map was unavailable and not used.

Our method delivers promising fidelity on all three testing datasets. First, our method is capable of adapting to different mesh topologies. For example, meshes in the Triplegangers dataset has denser vertices in the front face than behind while meshes in FaceScape have more uniform density, but our trained-once model performs equally well in both testing datasets. Second, our method does not sacrifice fidelity when input normal is unavailable during testing. Thanks to explicit geometry volume and robust model, pixels in the albedo map are precisely mapped onto the surface and therefore, yield no noise in rendering.

Finally, we acquired and tested our pre-trained model over the Generated Photos dataset containing only low-frequency albedo and normal map in Fig. 20. The dataset is generated by some anonymous method with only a single-view online image as input. Our method not only outputs a clear silhouette but also shows realistic pre-level detail when high-frequency input is unavailable.

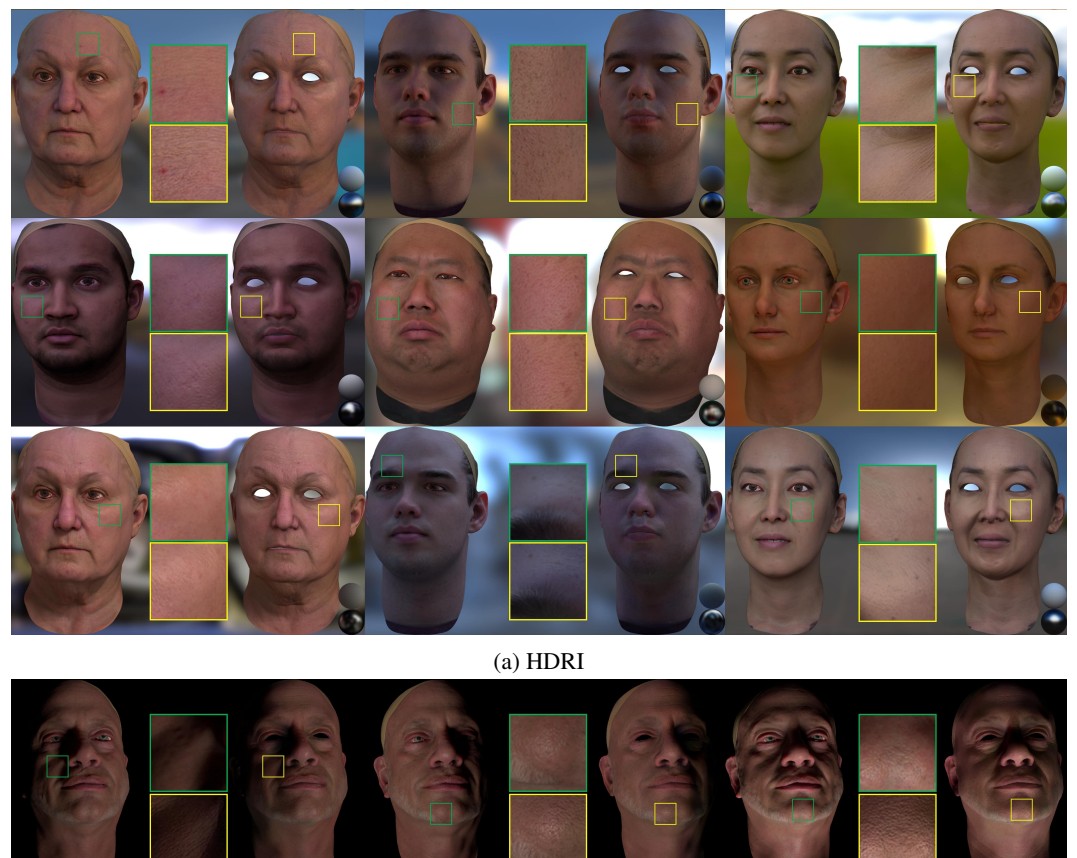

(a) HDRI

(b) OLAT

Figure 17: We show more qualitative comparisons of our method and Maya under (a) HDRI and (b) OLAT, with zoom-in on the images.

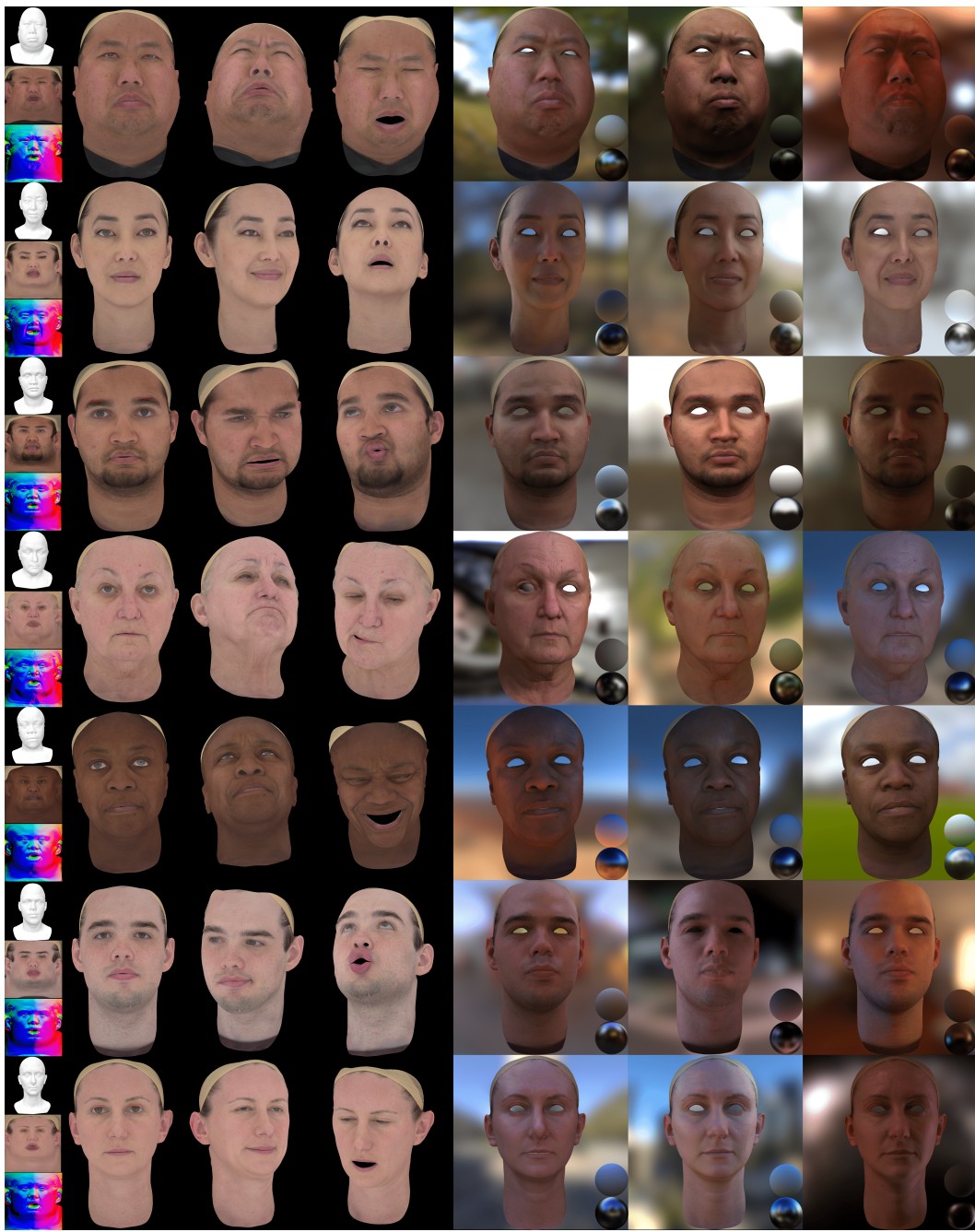

Figure 18: **Qualitative results.** In each row, We present our input (col. 1), three novel view rendering with neutral expression (col. 2), two other expressions under white illumination (col. 3, 4), and three frontal view rendering with neutral expression under different HDRI illumination (col. 5, 6, 7). Geometry, albedo map and normal map in UV space are list in the first column from top to bottom.

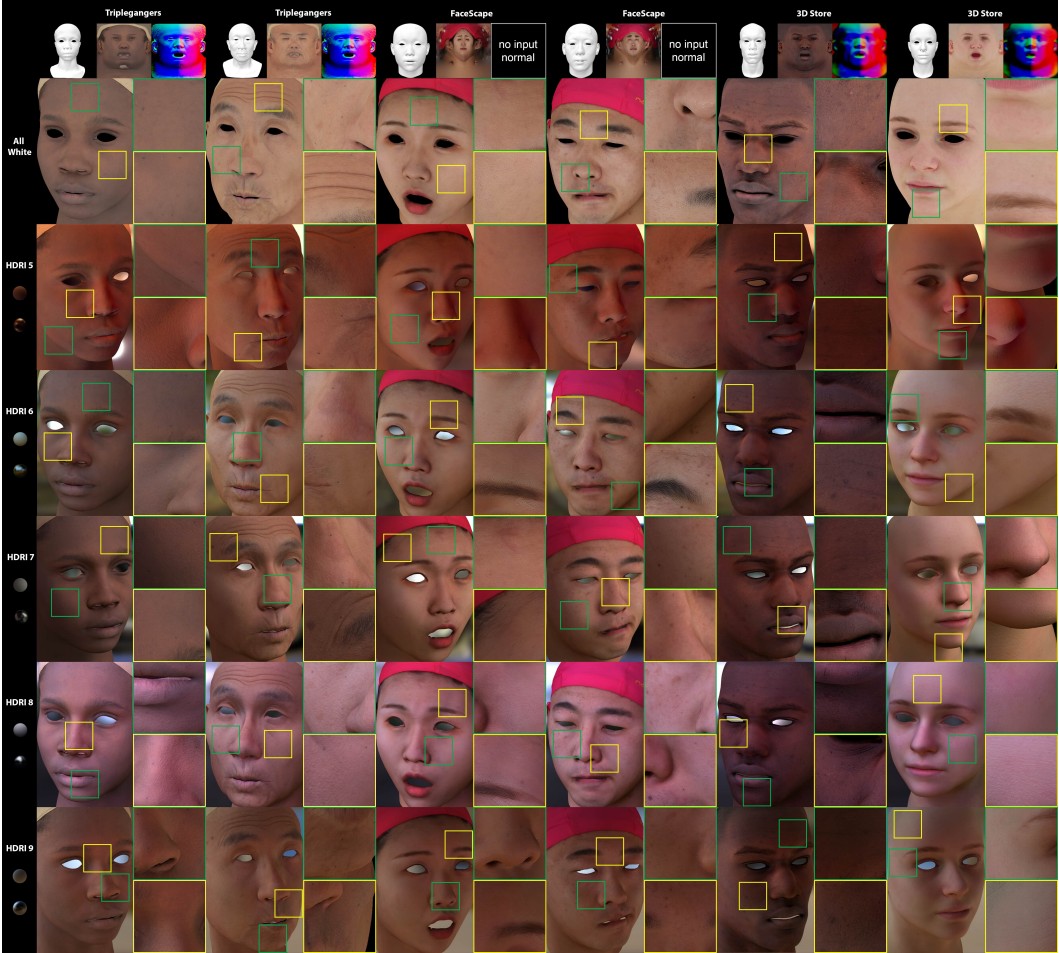

Figure 19: Testing performance of our trained-once models on other open source datasets. To demonstrate fidelity and generalization of our method, we further test our pre-trained model on Triplegangers, FaceScape and 3D Store datasets. Besides accurate shading, our rendering illustrates sharp detail in pore, wrinkle and hair.

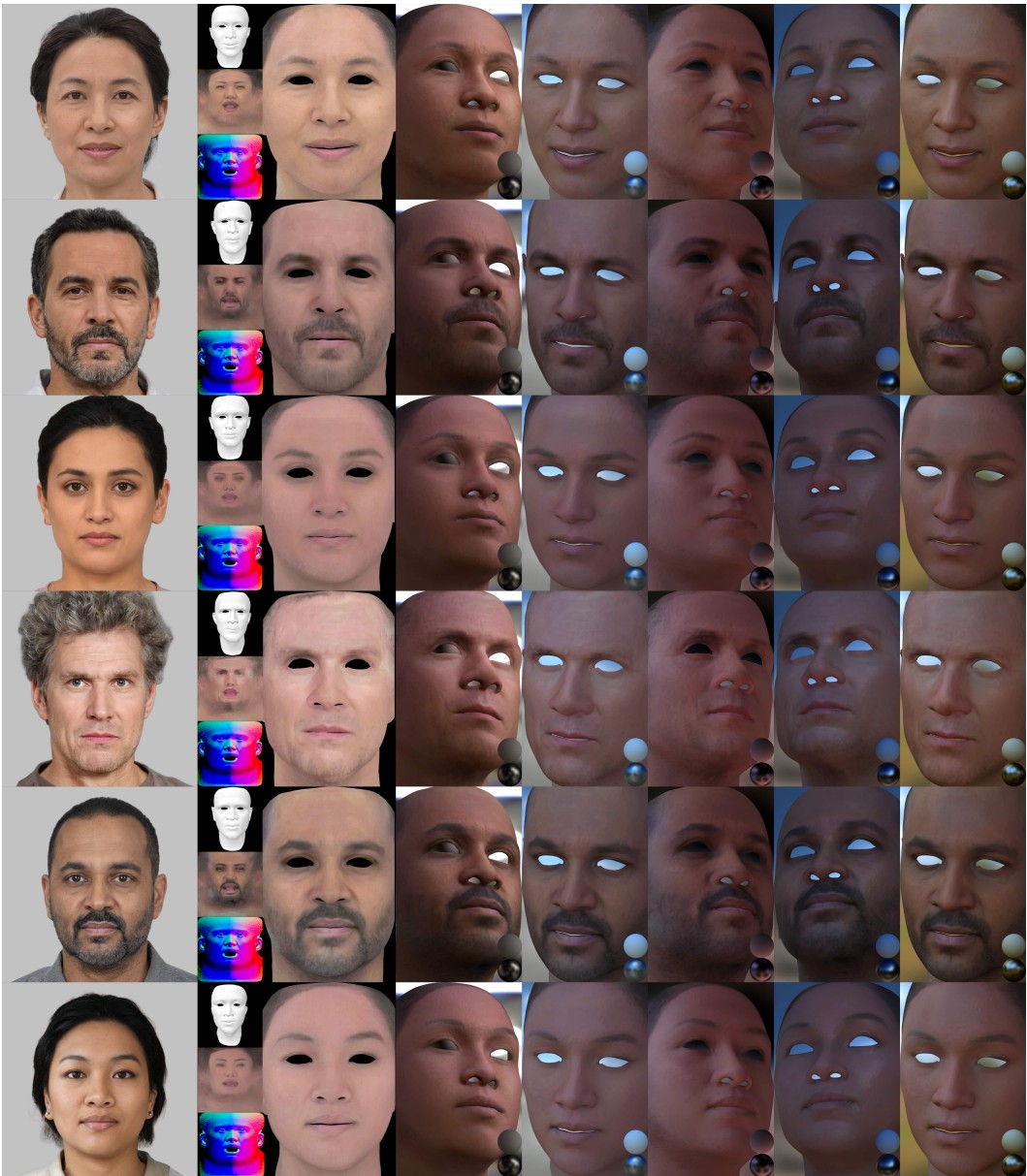

Figure 20: Testing performance of our trained-once models on GeneratedPhotos dataset (in-the-wild). Our model demonstrates high fidelity when accurate scans of albedo and normal map are unavailable in this test.

