# OpenReview forum: "Light Sampling Field and BRDF Representation for Physically-based Neural Rendering"
_ICLR.cc/2023/Conference — ICLR 2023 poster_

### Official Review · Reviewer_NfuH · 2022-10-20

**Confidence:** 4
**Correctness:** 4
**Technical Novelty And Significance:** 3
**Empirical Novelty And Significance:** 3
**Recommendation:** 6

**Clarity, Quality, Novelty And Reproducibility:**

Clarity: The paper is well-organized.

Quality: The experimental evaluation is adequate, and the results convincingly support the main claims.

Novelty: Good: The paper makes non-trivial advances over the current state-of-the-art.

Reproducibility: Key resources (e.g., proofs, code, data) are available, and key details (e.g., proofs, experimental setup) are sufficiently well-described for competent researchers to confidently reproduce the main results.


**Strength And Weaknesses:**

Strengths
+ There are some innovations in the method
1. It is an incremental innovation to represent previous discrete precomputed lighting points as a continuous MLP because many physical attributes have been modeled using MLPs since the publication of NeRF. (Eqn. 4).
2. It is novel to model the face's oil layer, epidermis, and dermis using a composited BRDF.
+ Relatively reliable experiments
1. Thorough experiments and evaluations on different modalities. The effects of different sampling strategies and rendering models are verified in 4.2 and 4.3, respectively.
2. As shown in Fig.10 and Tab.1, the method in this paper outperforms SOTAs in both qualitative and quantitative indicators, especially the rendering of illumination and shadows on the skin surface.
3. A large number of experiments are also given in the supplementary material to demonstrate the robustness of the proposed method under different parameters, such as illumination, resolution, etc.
+ Well written

Weaknesses
- Some parameters and method details are not clear
1. What is the ‘lighting embedding $z^l$’, and how to obtain it? It is suggested to give more details of the 'lighting embeddings $z^l$' in the first paragraph of page 5.
2. How to get the local SH representation in the paragraph of ‘ Light Sampling Field for Indirect Illumination.’? The authors should give an equation or a more detailed explanation.
3. The introduction of the proposed network structure, such as Material and Light Sampling Field Network, should be more detailed.
- What is the purpose of the ‘High-frequency Surface Normal ’experiment in the first paragraph of section 4.2? It is difficult to connect this experiment with the paper's key contributions.
- Eyes are indispensable components of the human face. However, none of the experiments provides results of them. Maybe the method may fail in these areas. It is also suggested to provide results or analysis here.


**Summary Of The Paper:**

This paper proposed a novel human-face neural rendering representation for physically-based rendering. The authors first give a volumetric lighting representation that accurately encodes the direct and indirect illumination positionally and dynamically gives an environment map. In addition, they also proposed a BRDF measurement representation that supports the PBR process by modeling specular, diffuse, and subsurface scattering for the human face. Finally, based on the above representations, the authors proposed a novel and lightweight neural PBR face shader that takes facial skin assets and environment maps (HDRIs) as input and efficiently renders photo-realistic, high-fidelity, and accurate images. The experiments show that the proposed PBR face shader significantly outperforms the state-of-the-art neural face rendering approaches.

**Summary Of The Review:**

Based on the novelty and reliable experiments, I suggest this paper be weakly accepted.

---

> ### Author Response · Authors · 2022-11-19
> **Response to Reviewer NfuH**
>
> We thank the reviewer for the constructive feedback. We added the missing references and provided more details, explanations about methods and data (e.g., real data capture and OLAT) in the revision for better understanding. We will freshen the memory of reviewer with the novelty and highlights of this paper and then address the concerns of the reviewer.
>
> **Novelty** Our paper proposes novel lighting and material representations for physically-based neural rendering, which makes the photo-realistic rendering of translucent objects (e.g., human skin, fruits) using networks possible. In particular, we separately model direct illumination and indirect illumination of the environment. For direct illumination, We use light importance sampling to simulate external light sources, such as light bulbs. And we implement ray tracing for the specular reflectance effect. For indirect illumination, we introduce a light sampling field that models location-aware illumination using SH. This learned local light-probe models subsurface scattering and inter-reflectance effects. Correspondingly, we model the material as a surface specular component, a diffuse surface component, and a subsurface scattering component. We evaluate our models on tasks of neural face shader and an inverse-rendering of the human face images. Comprehensive experiments show the effectiveness of our proposed representation and significant quality improvement on target tasks using our model.
>
> ## What is Lighting Embedding?
> Lighting embedding stores all pixel directions and intensities of a downsampled HDRI map (when mapped to a skydome) with a resolution of 100 * 150. We use this lighting embedding as input for the light Sampling Field to infer location-aware local illumination with SH.
>
> ## How to get local SH representation in indirect illumination?
> At each sampled location, we compute the local spherical harmonics by multiplying fixed orthonormalized Laplace’s spherical harmonics basis with predicted spherical harmonics parameters. Specifically, we use degree l=1 spherical harmonics to model indirect illumination for each color channel (i.e., red, green, blue). Therefore, we acquire 3 (number of color channels) x 4 (number of basis) = 12-dimension vector as local SH representation.
>
> ## What’s the purpose of High-frequency Surface Normal in experiments?
> Our original design of this experiment is to show the efficiency and correctness of our volume casting method in neural face shader. Even with high-frequency geometry information, we are able to map these details to the density field faithfully using our proposed sampling method. We agree with the reviewer that this experiment has less relevance to the main paper, and we move this to the appendix in the revision.
>
> ## How do we deal with eyes?
> In the application of neural face shader, we plan to capture and build an eye database that models the geometry and texture with real captured or synthetically rendered eye images. We will use our proposed model to train an eye-shader. Combined with a neural face shader, we have the complete rendering solution of the whole face. In the application of inverse-rendering (Fig. 8 and Fig. 10(a) last column), the input images have eyes so that the relighting is directly applied to the whole face images.

---

### Official Review · Reviewer_rwHM · 2022-10-22

**Confidence:** 4
**Correctness:** 3
**Technical Novelty And Significance:** 4
**Empirical Novelty And Significance:** 4
**Recommendation:** 8

**Clarity, Quality, Novelty And Reproducibility:**

Clarity: paper is well written, figures are visually appealing, but the notations and assumptions can be stated more clearly. Please refer to weakness.

Reproducibility: many of the implementation details are missing, making it hard for readers to reproduce.

**Strength And Weaknesses:**

Strength

- The proposed light sampling field is very novel and interesting to me. Method-wise, it is the first to model all the spatial-varying in-direct illumination in a coordinate-based MLP network. Speed-wise, it outputs the SH for fast and efficient rendering. Performance-wise, it generalizes well on novel subjects and illumination maps.

Weaknesses

- How is the material network generalized to different people? The input to material-net is only 3D position and ray direction, there is no person-specific information input to the network. So how can the material network generalize to different people with different skin properties?
Besides, the skin properties may change spatially even for the same person with different expressions.
- From the material network, the paper assumes that the specular strength and skin scattering are spatially identical across different subjects. If it is the case, the paper should mention this assumption in the paper. And I also feel that this assumption is a bit unrealistic.
- In eq(2), what is the $\omega$ stand for? View direction or light direction? It is very confusing that the notation is inconsistent between eq(2) and eq(3),(4).
- How is the 3D position x defined? It seems that it is defined in a head-centric coordinates system. Please provide details of it.
- In Figure 6, 2rd row, last 3 columns, it seems that the direct diffuse and indirect diffuse are somehow inconsistent with the 1st row. Is it a mistake? If not, can the author explain more? Why the direct diffuse here look more like indirect diffuse?

**Summary Of The Paper:**

This paper proposed a volumetric lighting representation that aims to model the indirect illumination given an illuminations map via a coordinate-based MLP. The proposed lighting representation achieves the state-of-the-art rendering quality on human faces, while being orders of magnitudes faster than the rendering engine.
In addition, the paper proposes a material network that explicitly models subsurface scattering on human face.
Both synthetic and real experiments demonstrate the render quality of the method outperforms prior works.

**Summary Of The Review:**

Overall, the paper proposed a novel approach for volumetric lighting representation. Experiments show that the volumetric lighting representation achieves state-of-the-art performance on face rendering and is 47-49 times faster than the industry-level rendering engine.
But, some assumptions and details are not clearly stated in the paper. Please address the questions and concerns in the weaknesses part.

---

### Official Review · Reviewer_oe43 · 2022-10-25

**Confidence:** 3
**Correctness:** 4
**Technical Novelty And Significance:** 4
**Empirical Novelty And Significance:** Not applicable
**Recommendation:** 8

**Clarity, Quality, Novelty And Reproducibility:**

Paper is well-written.
Novelty: modelling indirect lighting is already solved in traditional PBR(ray-tracing) but not addressed properly in neural rendering presentation.

Modelling sub-surface scattering is also new.


**Strength And Weaknesses:**

S:
The paper proposes to model light into direct and indirect lighting which takes local lighting such as inter-reflection into consideration.

Instead of only modelling surface BRDF, the proposed method also takes sub-surface into consideration which also enables it to generate more realistic results.

The proposed network is lightweight and achieves much better results compared with existing methods

W:
I have no major objection to the paper

Minor:
Page 4, Figure 3: (a) The Light Sampling Field network applies  (b) The Light Sampling Field network applies


**Summary Of The Paper:**

The paper presents a physical-based neural rendering network. The network codes lights and BRDF using 2 subnet-light sampling field network and material network. Then they are feed to construct light transport to generate radiance.

**Summary Of The Review:**

Overall good paper and I vote for acceptance initially.

---

### Official Review · Reviewer_M78E · 2022-10-25

**Confidence:** 4
**Correctness:** 2
**Technical Novelty And Significance:** 3
**Empirical Novelty And Significance:** 3
**Recommendation:** 3

**Clarity, Quality, Novelty And Reproducibility:**

Clarity: The methods and experiments section may lack important details. Some design choices are confusing. Please refer to weaknesses part for more details.

Quality: While authors show many quantitative and qualitative experiments, I may suggest that some experiments lack details or may not be a fair comparison. Therefore, it may be difficult to decide the quality of the proposed method. Please refer to weaknesses for more details. I believe C.1 is the really important experiments for this paper. It should be moved into the main paper and discussed in details.

Novelty: This paper targets at a challenging and important problem. The idea of separating surface reflection and scattering is novel and may be important. Authors also show that the method can significantly accelerate industrial rendering pipeline and can potentially be extended to objects other than faces. Therefore, I believe the paper is novel enough.

Reproducibility: Some current design choices in the method section may not be reasonable. Further explanation is needed to make it reproducible. Please refer to weaknesses section for more details.

**Details Of Ethics Concerns:**

I don't have any ethics concerns.

**Strength And Weaknesses:**

Strength of the paper:
1. The paper solves an important and challenging problem: building a neural renderer that can render complex translucent materials efficiently under environment map illumination. Some design choices may inspire future research, such as explicitly separating surface reflection and scattering.
2. Experiments show that the proposed neural shader can render high-quality human face images with much less time compared to an industrial standard path tracer. Authors also show that the proposed method may have the potential to be extended to handle general objects.

Weaknesses of the paper:
I list my major questions towards the paper below, which I hope authors can address in the rebuttal.

1. While I am very excited after reading the abstract and introduction, I find the method section quite confusing. I may suggest some necessary details are missing and some design choices may not be reasonable.

    a. Page 4 -- important light sampling: It is quite standard in computer graphics to do important sampling for the environment map in an unbiased way, why not just follow the standard method. In addition, the proposed sampling strategy seems to be problematic: why ignoring the textureless regions? Suppose the environment map is a cloudy sky, does that mean we won't have no samples for the sky region at all? In addition, why thresholding the intensity instead of important sampling. Won't that cause the method to miss important ambient lighting?

    b. Figure 3 -- material sampling field network: It is unexpected to encode ray direction omega when predicting material parameters. Doesn't that mean the material is angularly varying? It doesn't seem to be physically correct.

    c. Page 4, Figure 3 -- light sampling field: Similarly, if the output of the light sampling field is spherical harmonics, I assume it should be a radiance field covering lighting emitted in every direction. Why do we encode the ray direction as an input to the network? The SH should only be decided by the 3D location. In addition, how many orders of SH coefficients are used here?

    d. Page 4 -- indirect illumination: The term indirection illumination is not accurately used here. Indirect illumination includes both interreflection as well as light coming inside skin. Here authors clearly only model light coming through skin but not interreflection. This should be made clear in the paper.

    f. Page 5 -- dynamic objects: It is not accurate to claim that the method can handle dynamic object. Here the precomputed light lobe can only partially handle "moving" objects. If the object has some non-rigid transformations, the precomputed light lobe won't be able to handle the changing shadows and interreflection, etc. This should be explained clearly in the paper.

    g. Page 6 -- volume casting: Why proposing new volume casting scheme instead of following prior works, such as NeuS? In addition, the current volume casting method can be problematic as it uses distance between x and the intersection point to define density function, which means the same point on different rays may have very different behavior. I am not sure if that's desirable.

    h. Page 6, Eq. (4) -- material scattering: This part may lack important details: how to get e, \rou_x^s and \rou_x^ss for real data?

2. The focus of the experiments might be wrong. If I understand the method correctly, here authors use ground truth lighting and some material parameters to train the neural shader. Therefore, the proposed method should not be compared with prior neural reconstruction methods such as Neural-PIL and NeLF as the inputs are very different: they do not have GT material parameters and lighting as inputs. Instead, it should be compared with standard rendering pipeline to show that the proposed method can render realistic images with much less time consumption. I believe experiment in supplementary C.1 is the most important one and should be moved into the main paper with detailed discussion.

3. Further questions with respect to the experiments:

    a. Page 6 -- high-frequency surface normal: how to compute the high-frequency normal maps for training.

    b. Page 6 -- OLAT: the method is designed for using environment map lighting as input, how to handle OLAT iinput?

    c. Page 7 -- Light sampling field: this part is very confusing. The output of indirect light field is also SH. Why will the pre-calculated SH lead to worse results. In addition, I am curious if the implementation is correct in FIgure 5 (b), because I believe SH may result in missing specular highlight but should not cause global shift of the color.

    d. Page 7 -- Lighting model evaluation: This experiment may not be very meaningful as prior method PhySG and Neural-PIL design their method to jointly reconstruct environment while the proposed method has GT environment map as input. It is expected that sampling GT environment map should lead to more accurate lighting.

In the following, I also list my minor questions, which will not significantly influence my final ratings towards the paper. Authors can choose whether to address or not in the rebuttal.

1. Related work: when discussing portrait and face relighting, the series of works on Codec Avatar are missing, such as Ma et al. Pixel Codec Avatar and Bi et al. Deep Relightable Appearance Models for Animatable Faces. These works should be cited and discussed in related works.

**Summary Of The Paper:**

This paper proposes a physically-based neural shader that has the potential to model several challenging light transport effects realistically and efficiently. Authors designed several new modules to solve this challenging problem, including a neural lighting representation that separates direct and indirect illumination, a neural material model that handles surface reflection and subsurface scattering. Combined with neural volume rendering, authors show through experiments that their new neural shader can render transculent materials realistically, such as human skin.

**Summary Of The Review:**

This paper solves a challenging and important problem: how to design general neural renderer that can efficiently render complex materials, i.e. translucent materials realistically. The paper showed some interesting results on human face rendering and can significantly accelerate industrial standard software. However, I believe the current version may have many issues to be addressed. Many design choices in method section may be confusing and not reasonable. Experiment part may need to be re-designed to emphasize the major advantages of the proposed method. Therefore, I currently vote for rejecting this paper. Authors can consider solving my questions in the weaknesses section in the rebuttal. I will be happy to change my rating if authors can convince me. Thanks a lot!

---

### Decision · Program_Chairs · 2023-01-20

**Decision:**

Accept: poster

**Justification For Why Not Higher Score:**

Not higher because the topic is in a relatively specialized subfield, and because of the weaknesses listed above and in the reviews.

**Justification For Why Not Lower Score:**

There is a useful contribution here, and there is a possibility that future work will build on this paper, so it is of value to the community to publish it now, and at this venue.

**Metareview: Summary, Strengths And Weaknesses:**

All reviewers agree that the paper proposes a technique which can render high-quality human face images with less time compared to an industrial standard path tracer.  It was also agreed that the paper is likely to inspire follow-up work.

Conversely, it is clear from the rebuttal that the comparisons to Neural-PIL etc do indeed have the issue that different input data are available, hence these comparisons should be de-emphasized, and the difference in available data made very clear.

The Maya comparison is useful and should be presented as part of the main paper.

In addition, reviewers note a number of pipeline components (light sampling, volume casting) that differ from standard approaches.  It would be extremely valuable to include ablations that illustrate the effects of switching in and out these components.


**Note From Pc:**

if the above contains the word "oral" or "spotlight" please see: "oral" presentation means -> notable-top-5% and "spotlight" means -> notable-top-25%. As stated in our emails, we are disassociating presentation type from AC recommendations

**Summary Of Ac-Reviewer Meeting:**

The main topics of discussion were:
- given that some of the comparisons are not using equivalent training data, does the paper still make a useful contribution, and are the comparisons relevant.  The conclusion was that the contribution remains useful, that the Maya comparisons are useful, and ideally should be extended for any final version
- reviewers considered that the light sampling strategy was likely to lead to artefacts (e.g. a uniform sky), but that this criticism fell into the category of "we would have done this differently", which is not a reason to reject the paper as presented.  Conversely, there should be more discussion of this topic in the final version, and if the predicted artefacts are present, a suggestion made for their mitigation.
- similarly with the volume casting scheme - it should be shown visually what difference it makes if replaced with a more standard scheme.

All reviewers agreed verbally to accept as poster